# The c-Jun N-terminal kinase pathway of a vector insect is activated by virus capsid protein and promotes viral replication

**Wei Wang[1†], Wan Zhao[1†], Jing Li[1,2†], Lan Luo[1], Le Kang[1*], Feng Cui[1*]**

[1]State Key Laboratory of Integrated Management of Pest Insects and Rodents, Institute of Zoology, Chinese Academy of Sciences, Beijing, China; [2]University of Chinese Academy of Sciences, Beijing, China

**Abstract** No evidence has shown whether insect-borne viruses manipulate the c-Jun N-terminal kinase (JNK) signaling pathway of vector insects. Using a system comprising the plant virus *Rice stripe virus* (RSV) and its vector insect, the small brown planthopper, we have studied the response of the vector insect's JNK pathway to plant virus infection. We found that RSV increased the level of Tumor Necrosis Factor-$\alpha$ and decreased the level of G protein Pathway Suppressor 2 (GPS2) in the insect vector. The virus capsid protein competitively bound GPS2 to release it from inhibiting the JNK activation machinery. We confirmed that JNK activation promoted RSV replication in the vector, whereas JNK inhibition caused a significant reduction in virus production and thus delayed the disease incidence of plants. These findings suggest that inhibition of insect vector JNK may be a useful strategy for controling the transmission of plant viruses.

**\*For correspondence:** lkang@ioz. ac.cn (LK); cuif@ioz.ac.cn (FC)

[†]These authors contributed equally to this work

**Competing interests:** The authors declare that no competing interests exist.

## Introduction

Most plant viruses depend on sap-feeding insects from the order Hemiptera for their transmission (*Hogenhout et al., 2008*). The success and efficiency of virus transmission, especially for the persistent-propagative viruses, depend on specific interactions between the virus and proteins of the vector insects, allowing transport of the virus in and out of insect tissues and overcoming insect immune reactions for successful replication (*Blanc et al., 2014*). A detailed knowledge of the vector proteins that participate in viral transmission and replication could provide important clues for the development of control strategies to plant viruses.

*Rice stripe virus* (RSV), a member of the genus *Tenuivirus*, causes rice stripe disease, one of the most notorious rice diseases in temperate and subtropical East Asia (*Toriyama, 1986*). RSV is transmitted mainly by the small brown planthopper, *Laodelphax striatellus*, in a persistent, circulative-propagative manner (*Toriyama, 1986*). As with most persistent-propagative viruses, RSV cannot be transmitted directly between plants in the field, and there is a large difference in pathogenicity of the virus from the insect vectors and from the viruliferous plants (*Zhao et al., 2016b*). The viral genome consists of four single-stranded RNA segments and encodes seven proteins: RNA-dependent RNA polymerase, NS2 (RNA silencing suppressor), NSvc2 (putative membrane glycoprotein), NS3 (gene-silencing suppressor), CP (capsid protein), SP (nonstructural disease-specific protein), and NSvc4 (movement protein) (*Cui et al., 2016*; *Du et al., 2011*). CP is the most abundant viral protein and plays a crucial role in the retention and movement of the virus in the insect vector (*Blanc et al., 2014*).

Direct interaction between insect vector proteins and RSV proteins has been established, but knowledge about the functions of these vector proteins is limited. Numerous proteins from small brown planthopper interact with CP of RSV, such as atlasin, cuticular protein

**eLife digest** There are over a thousand different viruses that infect plants. Many plant viruses are transmitted by insects that feed on the plants, much as mosquitoes spread diseases between people when feeding on blood. Often the plant virus can replicate inside the cells of the insect. However, unlike in the plant hosts, the viruses do not seem to cause disease in the insects that carry them.

Rice stripe disease is a major viral disease of rice that can reduce the crop's yield by more than 50% in some areas. An insect called the small brown planthopper spreads the rice stripe virus between plants. Like other animals, insects have an immune system that protects them against viral infections. This means that the rice stripe virus must manipulate the planthopper's immune system in order to replicate inside the insect's cells. It was not clear how the virus did this, but answering this question could provide important clues to help scientists develop new ways to protect crops against plant viruses.

Wang, Zhao, Li et al. now show that rice stripe virus manipulates its insect host to produce more of a protein called TNF-α and less of a protein called GPS2. Moreover, a protein that makes up part of the virus also binds to GPS2. This stops GPS2 from inhibiting a conserved signaling pathway that involves an enzyme known as JNK. When the JNK signaling pathway becomes active, replication of the rice stripe virus inside the insect is accelerated. Further experiments showed that inhibiting JNK made it harder for the virus to replicate, which meant that it took longer for the disease to develop in rice plants.

These findings uncover a host of proteins that could be manipulated in insects to benefit rice agriculture. Such alterations could possibly be achieved through breeding or otherwise genetically modifying the insects to make them less able to carry viruses and then releasing them into wild populations. Alternatively, if further studies can identify chemicals that cause insect cells to alter the levels of the proteins, such chemicals could be administered to farmland to reduce the spread of viruses.

CPR1, jagunal, NAC domain protein, ribosomal proteins RPL5, RPL7a and RPL8, and vitellogenin (*Huo et al., 2014*; *Li et al., 2011*; *Liu et al., 2015*). There is evidence that cuticular protein CPR1 and vitellogenin, but not other vector proteins, affect RSV accumulation and transmission by binding with viral CP (*Huo et al., 2014*; *Liu et al., 2015*). RSV NS3 protein is able to attenuate the 26S proteasome- mediated host defense response by interacting directly with the planthopper RPN3 protein (*Xu et al., 2015*).

G protein pathway suppressor 2 (GPS2) is a small protein that was initially found to inhibit G protein (Ras)-activated MAPK (mitogen-activated protein kinase) signaling, especially JNK (c-Jun N-terminal kinase), in yeast and mammalian cells (*Spain et al., 1996*). The JNK signaling pathway can be activated by environmental stress and is implicated in multiple physiological processes, such as cell apoptosis/survival signaling, tumor development, diabetes, metabolism, embryonic development and aging (*Weston and Davis, 2002*). GPS2 in mammals is bifunctional. In the cell nucleus, GPS2 is a transcriptional cofactor that mediates both gene repression and activation as an intrinsic component of a major transcriptional repressor complex, the NcoR/SMRT nuclear receptor corepressor complex (*Zhang et al., 2002*). In the cytoplasm, GPS2 has a function in the regulation of JNK activation by inhibiting TRAF2/Ubc13 enzymatic activity in response to a proinflammatory cytokine, Tumor Necrosis Factor-α (TNF-α) (*Cardamone et al., 2012*). JNK is activated through phosphorylation of threonine and tyrosine residues within a Thr-Pro-Tyr motif located in kinase subdomain VIII. The activation of JNK requires the presence of the E2 ubiquitin-conjugating enzyme Ubc13, the assembly of K63-ubiquitin chains, and the integrity of the TRAF2 RING domain. GPS2 interacts with the TRAF2/Ubc13 K63 ubiquitylation machinery, translating into a significant decrease of JNK activation (*Cardamone et al., 2012*). The functions of GPS2 in invertebrates have not yet been determined.

Because viruses are obligatory cellular parasites, viral proteins can regulate host cellular signaling pathways. In mammals, hepatitis B virus (*Doria et al., 1995*), herpesvirus (*Jung and Desrosiers,*

*1995*), and human T-cell lymphotrophic virus (*Jin et al., 1997*) have been shown to affect Ras-dependent kinase cascades. In a compatible vector–virus relationship, such as vector insects and their transmitted plant or mammal viruses, how the viruses regulate the insect's Ras-dependent kinase cascadesand what benefits the viruses would get from this manipulation remain open questions. It has been reported that the JNK-like protein regulates phagocytosis and endocytosis in the *Aedes albopictus* mosquito cell line, C6/36, and West Nile virus infection may depend on JNK-mediated endocytosis (*Mizutani et al., 2003b*).

In this study, we explored the effect of RSV on the JNK signaling pathway in the virus's vector insect. We found that RSV activates the JNK signaling pathway in several ways, especially through the interaction of CP and GPS2, to secure a benefit in viral replication and disease outbreak in plants.

## Results

### Interaction between planthopper GPS2 and virus CP

To detect virus interactive proteins in the small brown planthoppers, we used RSV CP as bait to screen the insect cDNA library in a yeast two-hybrid system. A 1056 bp fragment encoding partial GPS2 was isolated on the SD/–Leu/–Trp/–His medium, but not on the SD/–Leu/–Trp/–His/–Ade medium. The interaction between CP and partial GPS2 was confirmed by further analysis with the yeast two-hybrid system. Yeast was co-transformed with the plasmids pDBLeu-CP and pDEST22-GPS2, and positive clones were obtained only on the SD/–Leu/–Trp/–His medium (*Figure 1A*), indicating a moderate strength of the interaction between CP and the partial GPS2.

The full-length GPS2 and RSV CP were recombinantly expressed in *Escherichia coli*, and the interaction between the two proteins was tested in vitro using a pull-down assay. The results showed that His-tagged GPS2 bound GST-fused CP, while no binding appeared in negative controls (*Figure 1B*). A co-immunoprecipitation assay using CP monoclonal antibody and recombinantly expressed His-CP pulled down GPS2 from the planthopper in vivo, consistent with an interaction between CP and GPS2 (*Figure 1C*).

### Features of the GPS2 sequence

After a BLAST search of the small brown planthopper transcriptome (*Zhao et al., 2016a*) with the partial *GPS2* sequence, the full-length open reading frame (ORF) of the planthopper *GPS2* was identified and cloned. Its 1299-bp nucleotides encode a 432 amino acid residue protein (GenBank accession KY435901). The *GPS2* partial sequence identified from the yeast two-hybrid was from nucleotide 277 to 1186 of the full-length ORF. The predicted molecular mass of the GPS2 protein was 47.8 kD, and the protein included an N-terminal coiled-coil region (nucleotides 73–336). Cellular localization of GPS2 was predicted to be both nuclear and cytoplasmic. Phylogenetic analysis of protein sequences supported the identity of the cloned *GPS2* gene because it clustered with mammalian and other insects' *GPS2* genes. Planthopper GPS2 was most similar to that of the termite *Zootermopsis nevadensis* (*Figure 1—figure supplement 1*). A paralogous gene, *GPS1* (GenBank accession KY435902), which clustered with mammalian *GPS1* gene (*Figure 1—figure supplement 1*), was also identified in the small brown planthopper transcriptome (*Zhao et al., 2016a*). Planthopper GPS1 and GPS2 had 17% amino acid sequence identity and GPS1 had no coiled-coil domain.

### Influence of RSV infection on the expression of *GPS2*

In order to clarify the influence of RSV infection on the expression of *GPS2*, the temporal and spatial expressions of *GPS2* were checked in non-viruliferous and viruliferous planthoppers using quantitative real-time PCR. In the non-viruliferous planthoppers, the transcript level of *GPS2* was highest in the eggs, was lowest in the middle nymph stages, and then increased until adult stage (*Figure 2A*). In adults, the *GPS2* transcript levels were highest in reproductive organs, testicle and ovary, and lowest in gut and fatbody (*Figure 2B*). The subcellular location of GPS2 was examined with immunohistochemistry in planthopper salivary gland cells (taking advantage of their large size) using anti-human GPS2 polyclonal antibody. This antibody had a high specificity for planthopper GPS2 (*Figure 2—figure supplement 1*). Immunofluorescence showed that GPS2 was located in both the nucleus and the cytoplasm (*Figure 2C*). When total protein was isolated from the nuclei

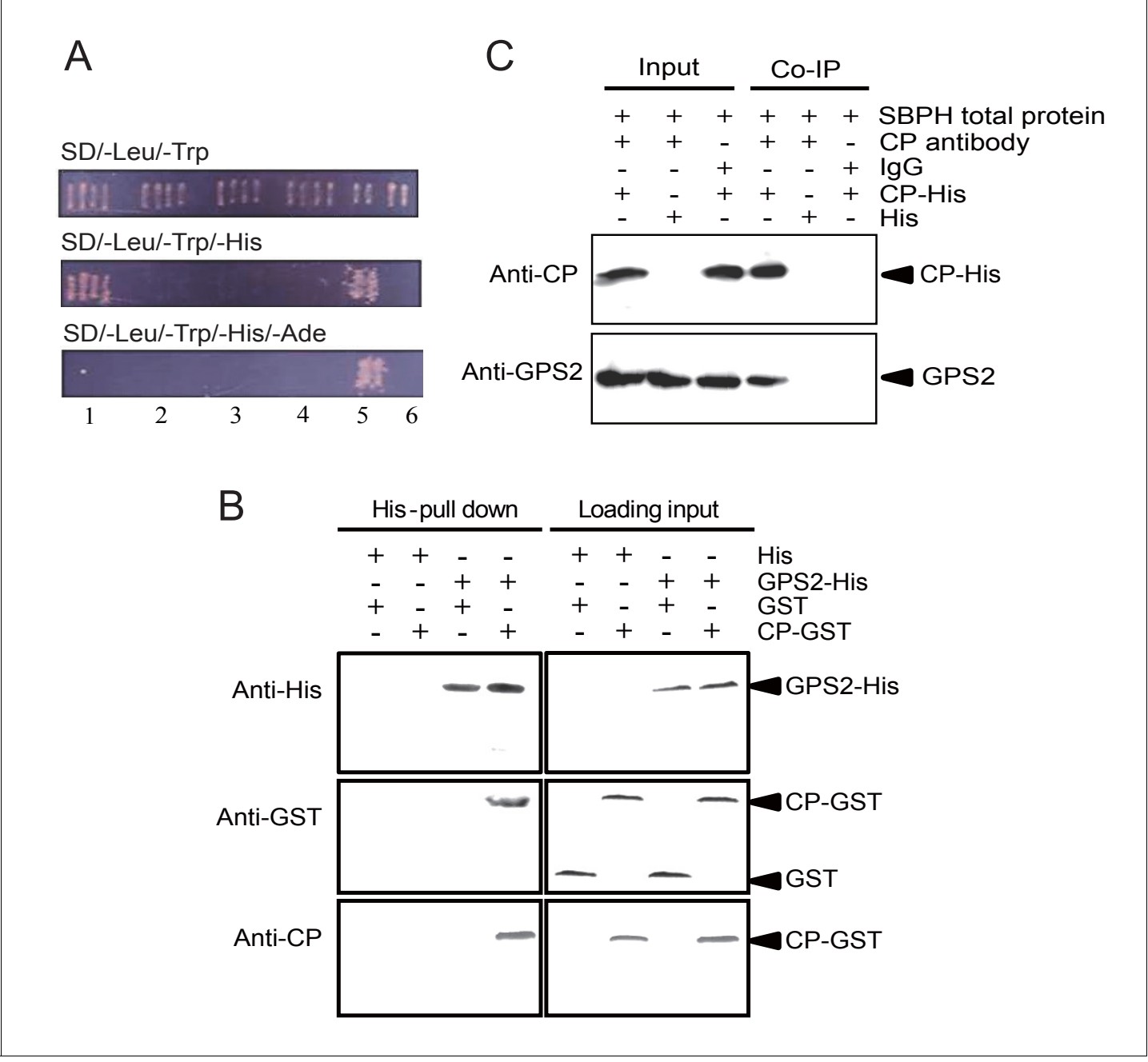

**Figure 1.** Small brown planthopper GPS2 binds *Rice Stripe Virus* capsid protein. (**A**) GPS2 binds CP in yeast two-hybrid assays. (1) pDBLeu-CP and pDEST22-GPS2; (2) pDBLeu-CP and pDEST22, self activation; (3) pDBLeu and pDEST22-GPS2, self activation; (4) pDBLeu and pDEST22, negative control; (5) pGBKT7-53 and pGADT7, positive control; (6) pGBKT7-Lam and pGADT7, negative control. Interaction between pDBLeu-CP and pDEST22-GPS2 was observed only on the SD/–Leu/–Trp/–His medium. (**B**) His-tag pull-down assay. Recombinantly expressed GPS2-His was bound to Ni Sepharose as a bait to hook the prey protein, recombinantly expressed CP-GST. The products from His vector (pET28a) and GST vector (pGEX-3X) were applied as negative controls. Anti-His, anti-GST, or anti-CP antibody was used to detect proteins. (**C**) Co-immunoprecipitation (CoIP) for recombinantly expressed CP-His and GPS2 from non-viruliferous planthopper total proteins. For (**B**) and (**C**), three independent biological replicates were carried out for each experiment.

The following figure supplement is available for figure 1:

**Figure supplement 1.** Phylogenetic tree of planthopper GPS2 and GPS1 proteins and those from other insect species.

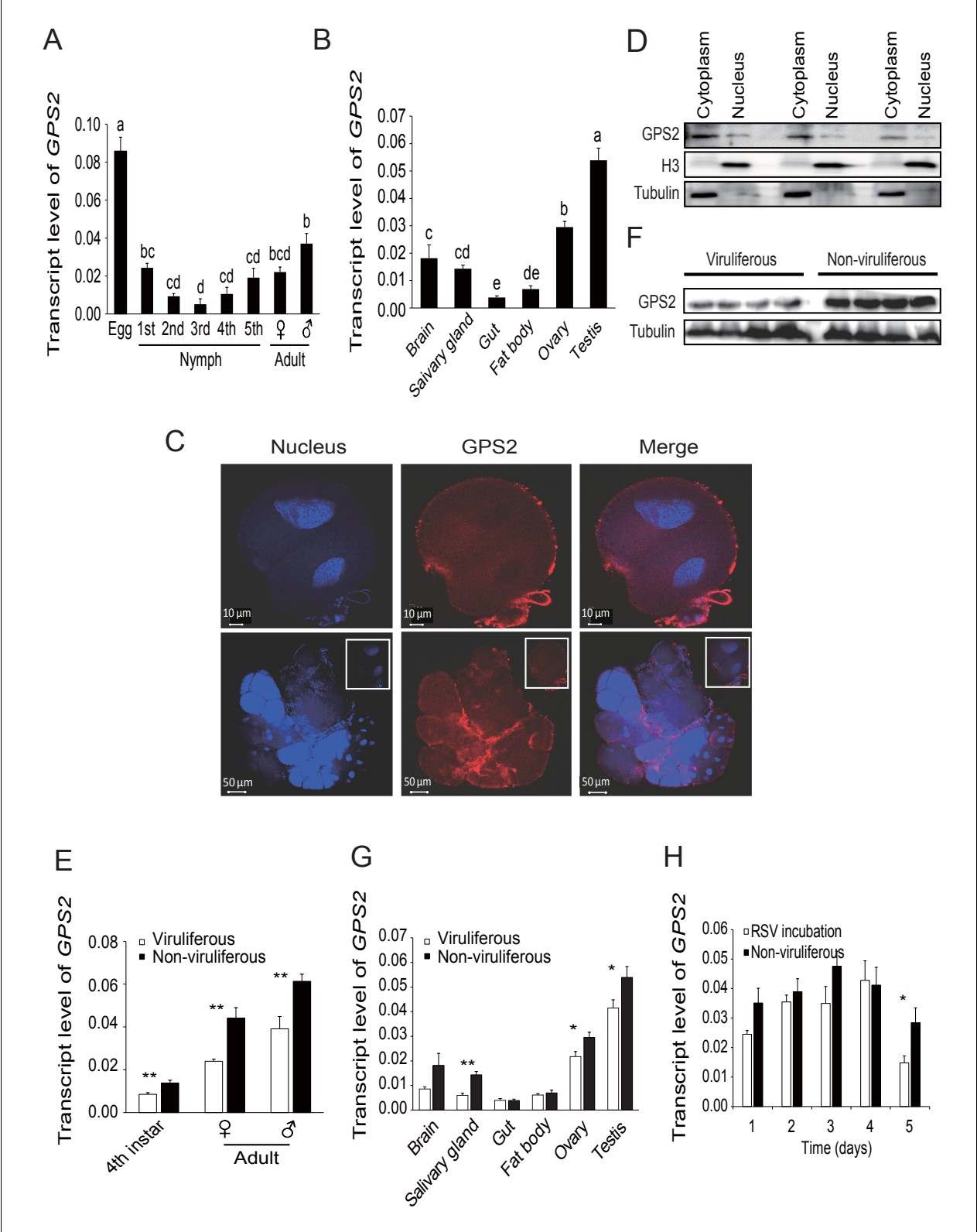

**Figure 2.** Temporal, spatial and subcellular expression of GPS2 in the planthopper. (A) Relative transcript levels of *GPS2* throughout the developmental stages of non-viruliferous planthoppers. (B) Relative transcript levels of *GPS2* in various tissues of non-viruliferous planthoppers. The transcript levels of *GPS2* are normalized to that of the *ef2* transcript. Different letters indicate significant differences in Tukey's multiple comparison test. (C) Subcellular localization of GPS2 in the salivary gland cells of non-viruliferous planthoppers revealed by an anti-human GPS2 polyclonal antibody in

*Figure 2 continued on next page*

*Figure 2 continued*

immunohistochemistry. Red is the positive signal. Nuclei are stained blue. (**D**) Western blotting showing GPS2 in the nuclei and cytoplasm of cells in insect whole-body samples. Reference proteins for nuclear and cytoplasmic proteins were histone H3 and tubulin, respectively, which were displayed using an anti-human H3 monoclonal antibody and an anti-human tubulin monoclonal antibody. (**E**) Comparison of relative transcript levels of *GPS2* between viruliferous and non-viruliferous fourth instar nymphs, females and males. (**F**) Western blotting showing GPS2 protein in fourth instar nymphs of viruliferous and non-viruliferous planthoppers. (**G**) Comparison of relative transcript levels of *GPS2* between the viruliferous and non-viruliferous various tissues. (**H**) Relative transcript levels of *GPS2* when RSV is incubated within the insects for five days. *p<0.05; **p<0.01.

The following source data and figure supplements are available for figure 2:

**Source data 1.** Numerical data that are represented as graphs in *Figure 2A,B,E,G,H*.
**Figure supplement 1.** Western blotting to show the specificity of the anti-human GPS2 polyclonal antibody for recognizing planthopper GPS2.
**Figure supplement 2.** Densitometry analysis for the GPS2 image bands from *Figure 2F*.

and the cytoplasm of insect whole bodies, more GPS2 protein was detected in the cytoplasm than in the nucleus (*Figure 2D*).

Compared to those in non-viruliferous planthoppers, the transcript level and protein level of *GPS2* in viruliferous insects was decreased in nymphs and adults (*Figure 2E,F*, *Figure 2—figure supplement 2*). The transcript levels of *GPS2* were lower in the salivary glands, ovaries, and testicles of viruliferous insects than of non-viruliferous insects (*Figure 2G*). When non-viruliferous insects were fed a diet containing RSV for 8 hr and the virus was allowed to incubate in insects for five days, the transcript level of *GPS2* decreased significantly on the fifth day (*Figure 2H*).

## GPS2 represses JNK activation in the planthoppers

To clarify whether GPS2 functions as a repressor of JNK activation in the planthoppers (as is the case in yeast and mammals [*Spain et al., 1996*]), JNK activation was investigated through the phosphorylation status of JNK when *GPS2* was knocked down by injection of double-strand RNAs. Two *JNK* genes were identified in the small brown planthopper transcriptome (*Zhao et al., 2016a*) using three human JNKs (GenBank accessions P45983, P45984, and P53779) as queries in BLASTp searches. The ORFs of the two putative *JNKs* were 1287 bp and 1125 bp, encoding a 49 kD protein (JNK1, GenBank accession KY435903) and a 43 kD protein (JNK2, GenBank accession KY435904), respectively. The amino acid identities of the two JNKs to the three JNKs of human were from 56% to 65% (*Figure 3—figure supplement 1*). The recombinantly expressed planthopper JNK1 and JNK2 were recognized by the anti-human JNK2 polyclonal antibody (*Figure 3—figure supplement 2*), and two bands corresponding to endogenous JNK1 and JNK2 appeared in planthopper samples using this antibody in western blotting (*Figure 3A*). When using an anti-phospho-human JNK2 antibody to show the phosphorylated JNK, only one band, apparently phosphorylated JNK1, was detected. The immune reaction decreased after the sample was treated with λ-phosphatase (*Figure 3A*).

A 219-bp double-strand RNA of *GPS2* was injected into viruliferous or non-viruliferous fourth instar planthoppers for transcript knockdown. The transcript knockdown for *GPS2* at the third day after injection was 87% and 91% in viruliferous and non-viruliferous insects, respectively. In both viruliferous and non-viruliferous insects, the phosphorylation of JNK was higher in the *GPS2* knockdown groups compared to the control groups, as revealed by the anti-phospho-human JNK2 antibody (*Figure 3B*, *Figure 3—figure supplement 3*), indicating that JNK was more significantly activated with less GPS2 and that GPS2 can repress JNK activation in the planthoppers.

## RSV activates the planthopper JNK signaling pathway

Considering that GPS2 represses JNK activation and that RSV's CP can bind GPS2, we examined whether RSV infection activated JNK signaling in the planthoppers. The phosphorylation level of JNK was compared between the viruliferous and non-viruliferous adult female, adult male, or fourth instar insects using the anti-phospho-human JNK2 antibody. A higher level of JNK phosphorylation was observed in the viruliferous instars (*Figure 4A*, *Figure 4—figure supplement 1A*), but not in viruliferous female or male adults (*Figure 4—figure supplement 2*). The most obvious increases in

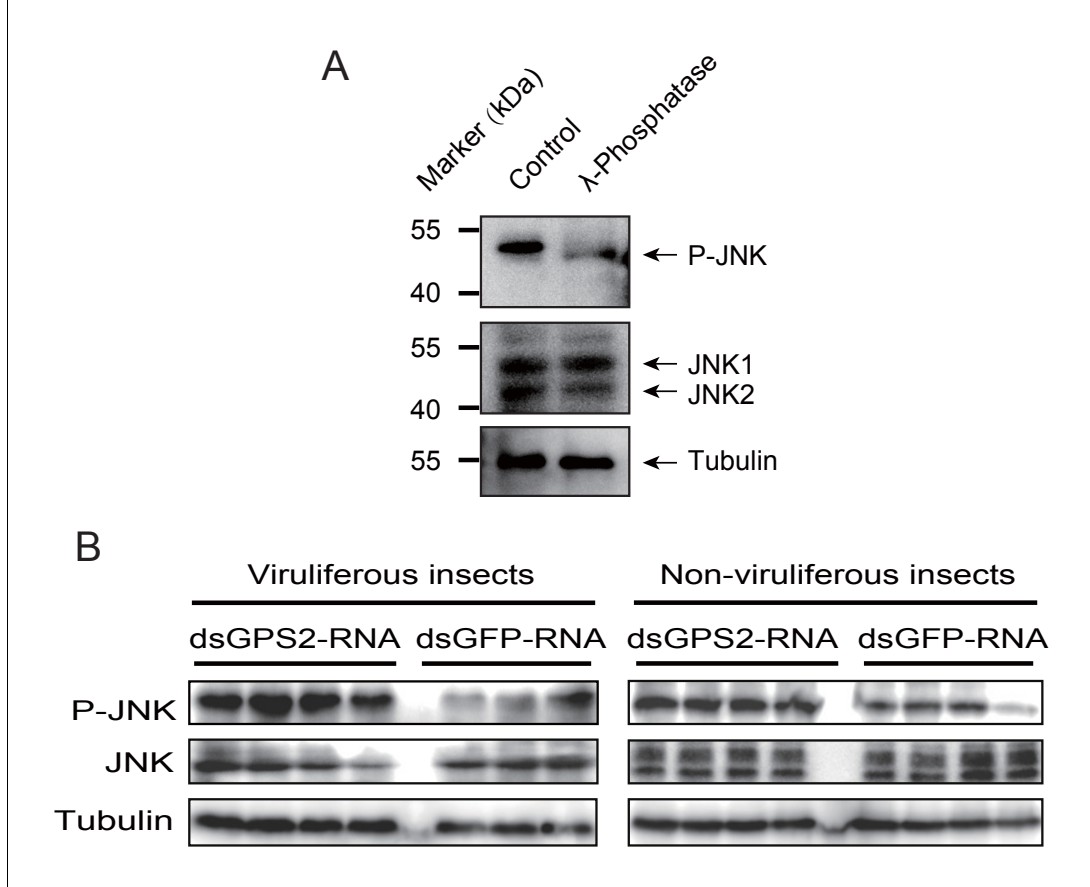

**Figure 3.** GPS2 represses JNK activation in the planthopper. (**A**) Western blot of the phosphorylated JNK (P-JNK) and total JNKs in non-viruliferous fourth instar planthoppers before and after the treatment with λ-phosphatase. Total protein was incubated with λ-phosphatase for 1 hr at 30°C. Three independent biological replicates were carried out. Here we show one representative result. (**B**) Western blotting showing P-JNK in viruliferous or non-viruliferous fourth instar nymphs when ds*GPS2*-RNA was injected. Levels were determined 3 d after injections. ds*GFP*–RNA injection was used as control. The P-JNK, total JNKs, and tubulin were detected using an anti-phospho-human JNK2 antibody, an anti-human JNK2 polyclonal antibody, and an anti-human tubulin monoclonal antibody, respectively.

The following figure supplements are available for figure 3:

**Figure supplement 1.** Amino acid sequence alignments of planthopper JNKs (LsJNK1 and LsJNK2) and human JNKs (HsJNK1, HsJNK2, HsJNK3).

**Figure supplement 2.** Recombinant expression of planthopper JNK1 and JNK2 for specificity verification of the anti-human JNK2 polyclonal antibody.

**Figure supplement 3.** Densitometry analysis for the phosphorylated JNK (P-JNK) image bands from *Figure 3B*.

JNK phosphorylation level appeared in the gut, salivary glands, ovary, and testicle of viruliferous insects compared to the non-viruliferous insects (*Figure 4B*, *Figure 4—figure supplement 1B–E*).

In many systems, TNF-α is known to be an upstream proinflammatory signaling molecule for JNK activation (*Grivennikov et al., 2010*). A putative *TNF-α* transcript (GenBank accession KY435905), encoding 389 amino acid residues, was identified from our small brown planthopper transcriptome (*Zhao et al., 2016a*). The encoded protein contains a conserved TNF domain, as do the human homolog (NP_000585) and the *Drosophila* homolog (also called Eiger) (NP_724878) (*Figure 4—figure supplement 3*). An enzyme linked immunosorbent assay using human TNF-α monoclonal antibody showed that the level of the TNF-α protein increased two-fold when insects were infected by RSV (*Figure 4C*). However, when the expression of *TNF-α* was lowered (70% knockdown of its transcript level), RSV did not upregulate the phosphorylation of JNK (*Figure 4D*, *Figure 4—figure*

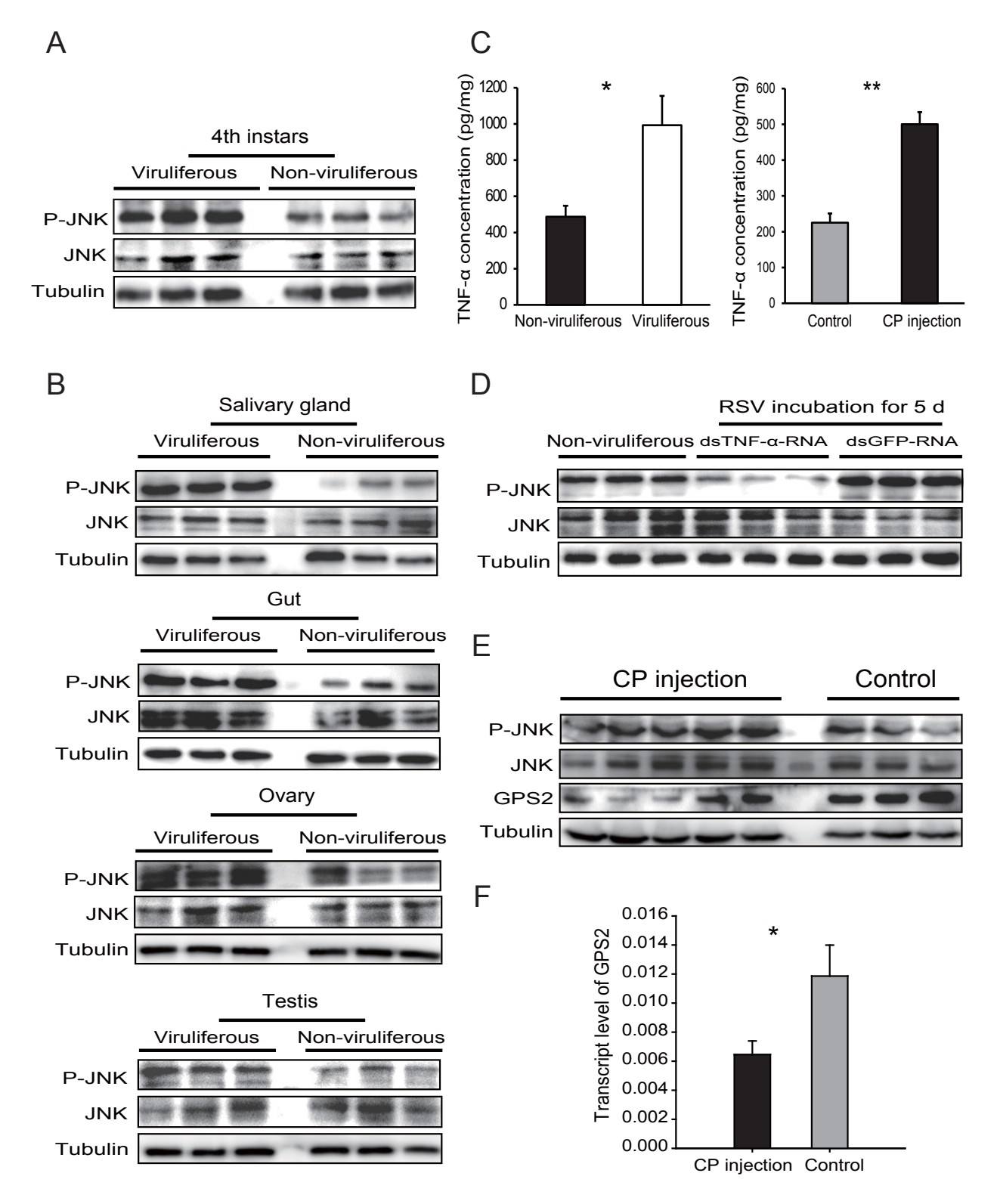

**Figure 4.** *Rice stripe virus* activates the JNK signaling pathway. (**A**) Western blotting showing the phosphorylated JNK (P-JNK) in the fourth instar nymphs of viruliferous and non-viruliferous planthoppers. (**B**) Western blotting showing the P-JNK in salivary gland, gut, ovary, and testis of viruliferous and non-viruliferous planthoppers. (**C**) The TNF-α levels in the fourth instar nymphs of viruliferous and non-viruliferous planthoppers, or in CP-His-protein-injected non-viruliferous insects and negative control groups, determined using a human TNF-α ELISA Kit. (**D**) Western blotting showing the

*Figure 4 continued on next page*

*Figure 4 continued*

P-JNK in the fourth instar nymphs when ds*TNF-α*-RNA was injected and RSV was incubated in insects for five days. (**E**) Western blotting showing the P-JNK and GPS2 in non-viruliferous fourth instar nymphs at 24 hr after CP-His protein injection. The control insects were injected with purified products from pET28a vector. (**F**) Relative transcript levels of *GPS2* after CP-His protein injection. The P-JNK, total JNK, GPS2, and tubulin were detected using an anti-phospho-human JNK2 polyclonal antibody, an anti-human JNK2 polyclonal antibody, an anti-human GPS2 polyclonal antibody, and an anti-human tubulin monoclonal antibody, respectively. *p<0.05; **p<0.01.

The following source data and figure supplements are available for figure 4:

**Source data 1.** Numerical data that are represented as graphs in *Figure 4C,F*.

**Figure supplement 1.** Densitometry analysis for the phosphorylated JNK (P-JNK) and GPS2 image bands from *Figures 4A, B, D and E*.

**Figure supplement 2.** Western blot of the phosphorylated JNK (P-JNK) in female and male viruliferous and non-viruliferous planthoppers.

**Figure supplement 3.** Protein characteristics of TNF-α from small brown planthopper, *Drosophila* fruit fly, and human.

*supplement 1F*). We conclude that RSV infection stimulated an inflammatory response and activated the JNK signaling pathway in the planthopper.

## RSV's CP activates the JNK signaling pathway in the planthoppers

To determine whether JNK activation was mediated by the capsid protein of RSV, recombinantly expressed and purified CP was injected into non-viruliferous fourth instar planthoppers. A higher level of JNK phosphorylation was observed in western blotting in insects at 24 hr after CP administration than in the control group, indicating that CP activated JNK (*Figure 4E*, *Figure 4—figure supplement 1G*). At the same time, the transcript level (*Figure 4F*) and the protein level of GPS2 (*Figure 4E*, *Figure 4—figure supplement 1H*) were significantly lower in the presence of CP. The TNF-α protein level increased 2.2-fold after CP administration (*Figure 4C*). Thus, regulation of the JNK pathway by RSV in the planthopper is caused by the virus's capsid protein.

## RSV's CP competes with Ubc13 in binding GPS2

Because the E2 ubiquitin-conjugating enzyme Ubc13 is part of the enzymatic machinery for JNK activation in mammals (*Cardamone et al., 2012*), we studied the function of Ubc13 in the JNK pathway in the planthopper. First, we found a putative *Ubc13* ORF from the small brown planthopper transcriptome. The planthopper *Ubc13* encoded a 17-kD protein (GenBank accession KY435906) and had 82% amino acid identity with the human homolog (NP_003339). The recombinantly expressed Ubc13 with a GST-tag was a 44.2-kD protein. Next, we knocked down the *Ubc13* transcript and tested the JNK activation using the anti-phospho-human JNK2 antibody and treatment withmouse TNF-α. The mouse TNF-α was able to stimulate JNK phosphorylation in the insects (*Figure 5—figure supplement 1*). The knockdown of the *Ubc13* transcript was 91%, and the extent of JNK phosphorylation was less than in the control group (*Figure 5A*, *Figure 5—figure supplement 2*). This means that *Ubc13* takes part in the JNK activation in planthoppers.

A pull-down experiment using three recombinantly expressed proteins showed that GPS2 was able to pull Ubc13 down (*Figure 5B*). However, when CP was present in different amounts, the binding between GPS2 and Ubc13 was affected, the more CP, the less Ubc13 binding to GPS2 occurred (*Figure 5B and C*). Recombinantly expressed GPS2 can pull down Ubc13 from an extract of non-viruliferous planthoppers. When the recombinantly expressed CP was loaded, less Ubc13 was pulled down from non-viruliferous planthoppers (*Figure 5D*). When an extract from non-viruliferous or viruliferous insects was passed over a Ni Sepharose column to which recombinantly expressed GPS2 was bound, much more Ubc13 was recovered from non-viruliferous than from viruliferous insects (*Figure 5E*). These results indicate that RSV's CP competes with Ubc13 in binding GPS2, thus releasing GPS2 from inhibiting the JNK pathway.

To verify whether the competition between CP and Ubc13 happened at a specific region of GPS2, five fragments (*Figure 5—figure supplement 3*) — N1 (nucleotides 1–465), N2 (1–657), N3 (1–903), N4 (466–885), and C (886–1296) of GPS2 — were expressed and tested for potential

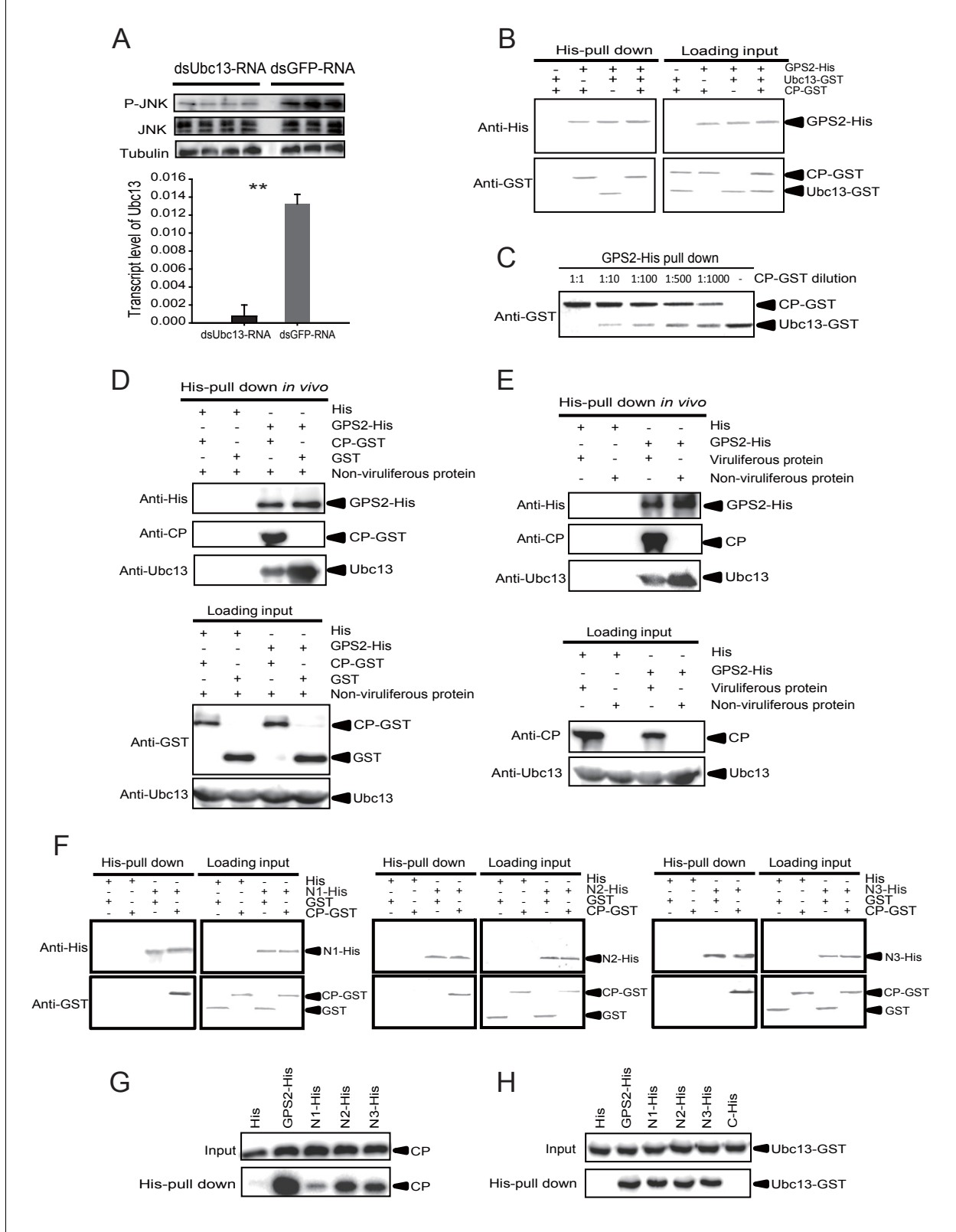

**Figure 5.** The *Rice stripe virus* capsid protein competes with Ubc13 in binding GPS2. (**A**) Western blotting showing the phosphorylated JNK (P-JNK) when ds*Ubc13*-RNA and mouse TNF-α were injected into the non-viruliferous fourth instar nymphs of the planthopper. Injections of ds*GFP*–RNA or 20 mM Tris-HCl containing 0.1% Tween 20 were used as controls. Relative transcript levels of *Ubc13* were compared after three days of treatment. **p<0.01. (**B**) and (**C**) His-tag pull-down assay for the competitive binding of GPS2 by CP and Ubc13 using recombinantly expressed proteins.

*Figure 5 continued on next page*

*Figure 5 continued*

Recombinantly expressed GPS2-His was bound to Ni Sepharose as a bait for hooking Ubc13-GST at a series of dilutions of CP-GST. (**D**) His-tag pull-down assay for the competitive binding of GPS2 by CP and in vivo Ubc13. In vitro expressed GPS2-His was bound to Ni Sepharose. CP-GST or GST was applied to the GPS2-His bound Sepharose. Then, the total proteins from non-viruliferous planthoppers were applied to the Sepharose. (**E**) His-tag pull-down assay for the competitive binding of GPS2 by in vivo CP and in vivo Ubc13. Recombinantly expressed GPS2-His was bound to Ni Sepharose. Then the total proteins from viruliferous or non-viruliferous planthoppers were applied to the Sepharose. (**F**) His-tag pull-down assay for the interaction between CP-GST and the N1-His (nucleotides 1–465), N2-His (1-657), or N3-His (1-903) fragment of GPS2. The expression products from His vector (pET28a) and GST vector (pGEX-3X) were applied as negative controls. (**G**) His-tag pull-down assay for the interaction between the N1-His, N2-His, or N3-His fragment of GPS2 and the CP from the total protein of viruliferous planthoppers. The expression product from His vector (pET28a) was applied as negative control. (**H**) His-tag pull-down assay for the interaction between the N1-His, N2-His, N3-His, or C-His (nucleotides 886–1296) fragment of GPS2 and Ubc13-GST. The expression product from His vector (pET28a) was applied as negative control. Anti-phospho-human JNK2 polyclonal antibody, anti-human JNK2 polyclonal antibody, anti-human tubulin monoclonal antibody, anti-human Ubc13 monoclonal antibody, anti-CP monoclonal, anti-His monoclonal, and anti-GST polyclonal antibody were used to detect proteins. From (**B**) to (**H**), three independent replicates were carried out for each experiment. We show one representative result for each experiment here.

The following source data and figure supplements are available for figure 5:

**Source data 1.** Numerical data that are represented as a graph in *Figure 5A*.

**Figure supplement 1.** Western blot of the phosphorylated JNK (P-JNK) in response to TNF-α treatment.

**Figure supplement 2.** Densitometry analysis for the phosphorylated JNK (P-JNK) image bands from *Figure 5A*.

**Figure supplement 3.** Positions of the five fragments, N1, N2, N3, N4, and C, within GPS2.

**Figure supplement 4.** His-tag pull down assay for the interactions of GPS2 fragments with CP or Ubc13.

interactions with CP or Ubc13 using pull-down assays. The results showed that N1, N2, and N3 were all able to pull down recombinantly expressed CP (*Figure 5F*) as well as the CP from viruliferous insects (*Figure 5G*). The same fragments were also able to pull down recombinantly expressed Ubc13 (*Figure 5H*). By contrast, neither fragment N4 nor fragment C bound either CP (*Figure 5—figure supplement 4A,B*) or Ubc13 (*Figure 5H*, *Figure 5—figure supplement 4C*). N1, N2, and N3 all contain the coiled-coil domain, whereas N4 does not, suggesting that the N-terminal coiled-coil region of GPS2 is the binding region for CP and Ubc13.

## JNK activation facilitates the proliferation of RSV in the planthoppers

To explore the effect of JNK activation on RSV transmission, we assessed virus proliferation in insects and the virus transmission efficiency when JNK was activated or inhibited. First, we knocked down the *GPS2* transcript by injection of ds*GPS2*-RNA to activate JNK. Compared to the control group (ds*GFP*-RNA injected), the transcript level of *GPS2* in the treatment group was reduced by 49% while the RNA level of *CP* increased by 18-fold (*Figure 6A and B*), demonstrating that RSV replicated more in insects with lower GPS2 amounts. Second, we lowered the *TNF-α* transcript level by injection of ds*TNF-α*-RNA to inhibit JNK. When the transcript level of *TNF-α* was knocked down by 70%, the RNA level of *CP* decreased by 80% (*Figure 6C*).

We also manipulated JNK directly to test the effect of JNK signaling on RSV proliferation in the planthoppers. When ds*JNK1*-RNA and ds*JNK2*-RNA were injected simultaneously into insects, the transcript levels of *JNK1* and *JNK2* were reduced by 65% and 82%, respectively (*Figure 6D*). There was 71% less *CP* RNA in the ds*JNKs*-RNA-injected insects than in the ds*GFP*-RNA-injected insects (*Figure 6E*). In pharmacological experiments, the JNK-specific chemical inhibitor SP600125 or the JNK agonist TNF-α from mouse was injected into the insects. The JNK phosphorylation level reduced by 62% after treatment with SP600125 in the small brown planthoppers (*Figure 6—figure supplement 1*), and the *CP* RNA level reduced by 28% (*Figure 6F*). By contrast, treatment with TNF-α increased the *CP* RNA level around two-fold with 50% increase of the JNK phosphorylation level (*Figure 6G*, *Figure 5—figure supplement 1*). Taken together, these results show that JNK activation facilitated the proliferation of RSV in the planthoppers.

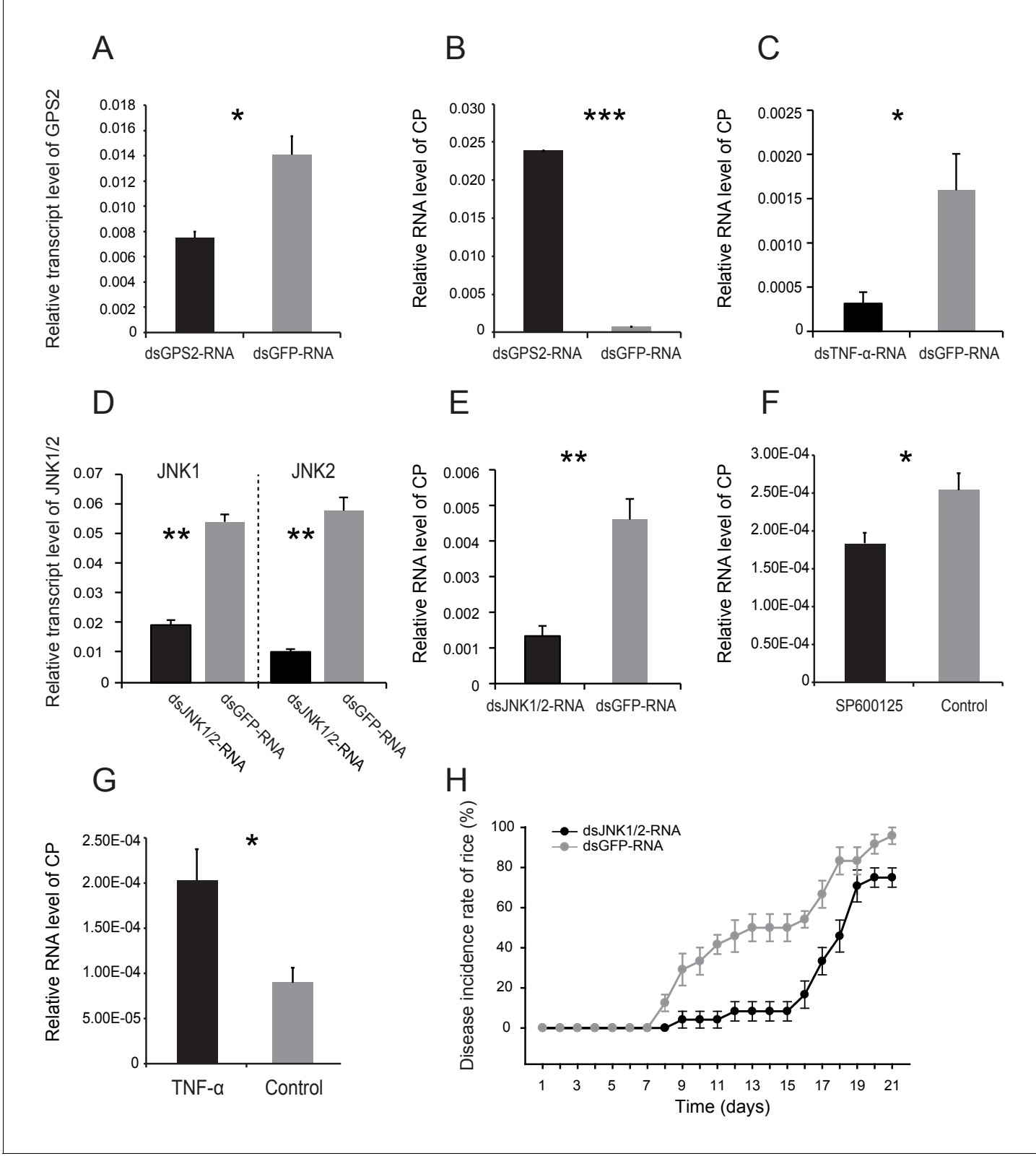

**Figure 6.** JNK activation facilitates the proliferation of *Rice stripe virus* in insects. (**A**) Relative transcript levels of *GPS2* 7 d after ds*GPS2*-RNA or ds*GFP*-RNA injection. (**B**) Relative RNA levels of RSV *CP* in ds*GPS2*-RNA-injected and ds*GFP*-RNA-injected planthoppers after 5 d of virus incubation. (**C**) Relative RNA levels of RSV *CP* in ds*TNF-α*-RNA-injected and ds*GFP*-RNA-injected planthoppers after 5 d of virus incubation. (**D**) Relative transcript levels of *JNK1* and *JNK2* 3 d after injection of the mixture of ds*JNK1*-RNA and ds*JNK2*-RNA. (**E**) Relative RNA levels of RSV *CP* in ds*JNK1/2*-RNA-

*Figure 6 continued on next page*

*Figure 6 continued*

injected and ds*GFP*-RNA-injected planthoppers after 5 d of virus incubation. (**F**) Relative RNA level of RSV *CP* in JNK-specific chemical inhibitor SP600125-injected planthoppers after 5 d of virus incubation. The control group was injected with an equal amount of 2% DMSO. (**G**) Relative RNA level of RSV *CP* in mouse TNF-α-injected planthoppers after 5 d of virus incubation. The control group was injected with an equal amount of 20 mM Tris-HCl containing 0.1% Tween 20. (**H**) The disease incidence rate of the rice plants fed upon by *JNK1* and *JNK2* knockdown planthoppers. The plants were placed at 26°C with 16 hr of light per day to observe disease symptoms. Six groups of plants per replicate and four replicates were used to calculate the disease incidence rate on each day. Plants fed upon by ds*GFP*-RNA injected insects were used as controls. *p<0.05; **p<0.01; ***p<0.001.

The following source data and figure supplements are available for figure 6:

**Source data 1.** Numerical data that are represented as graphs in *Figure 6A–G*.
**Figure supplement 1.** Western blot of phosphorylated JNK (P-JNK) after SP600125 treatment.
**Figure supplement 2.** The disease incidence rate of the rice plants fed upon by *GPS2* knockdown planthoppers.

## Effect of insect JNK pathway on the incidence of plant disease

To show the effect of manipulation of the JNK pathway of vector insects in rice plants, the disease incidence of rice plants was determined when the plants were fed upon by viruliferous insects in which the expressions of *GPS2* or *JNKs* were lowered. In the ds*GPS2*-RNA-injected insects, RSV replication increased (*Figure 6B*), but the disease incidence rates of the rice plants fed upon by the ds*GPS2*-RNA-injected insects were similar to those fed upon by the ds*GFP*-RNA-injected insects (*Figure 6—figure supplement 2*). When the levels of the two *JNK* transcripts were lowered in insects in which RSV replication was inhibited (*Figure 6E*), the disease incidence of plants was delayed (*Figure 6H*). Within 15 d, 50% of the plants in the control group showed disease symptoms whereas only 8% of plants in the *JNKs*-knocked-down group showed disease symptoms. However, 75% plants in the *JNKs*-knocked-down group showed disease symptom by 21 d, close to the 95% disease incidence rate in the control group. This suggests that inhibition of the JNK pathway retards the incidence of disease in plants by restricting RSV replication in the planthoppers.

## Discussion

Even in the very well-studied mosquito systems, little is known about how viruses that cause diseases in humans, animals or plants manipulate the cellular JNK signaling pathway of their vector insects. Our study clarifies for the first time that the virus enhances its replication in a vector insect by activating the vector's JNK pathway, and that JNK inhibition results in a delayed disease incidence in plants. We provide a model to summarize how RSV manipulates the upstream signal molecules and the downstream suppressor to activate the JNK pathway in the planthoppers (*Figure 7*).

Our finding that JNK activation by RSV facilitates virus replication in the small brown planthopper reflects a conservative evolution scenario in which the virus–vector and virus–host interactions both utilize the JNK pathway. The JNK signaling pathway in mammal or invertebrate hosts has been reported to be involved in or affected by infections by various viruses. For instance, infection with rhesus rotavirus, herpes simplex virus type 1, or varicella-zoster virus results in the activation of JNK in mammalian cells. Inhibition of JNK by SP600125 reduced virus replication (*Holloway and Coulson, 2006*; *McLean and Bachenheimer, 1999*; *Zapata et al., 2007*). In *Bombyx mori*, the magnitude and pattern of JNK activation were dependent on the multiplicity of nucleopolyhedrovirus infection, and inhibition of JNK reduced occlusion body formation and budded virus production (*Katsuma et al., 2007*). In a crustacean, *Litopenaeus vannamei*, JNK was activated in response to white spot syndrome virus infection and the virus proliferation benefited from JNK activation (*Shi et al., 2012*). Although this phenotype is common in virus–vector and virus–host systems, there should be some mechanisms in vector insects to limit the virus replication to a sustainable level.

Our study shows that the capsid protein of RSV not only induces a decrease of GPS2 expression, but also directly binds GPS2 so as to release GPS2 from repressing the JNK pathway. The N-terminal

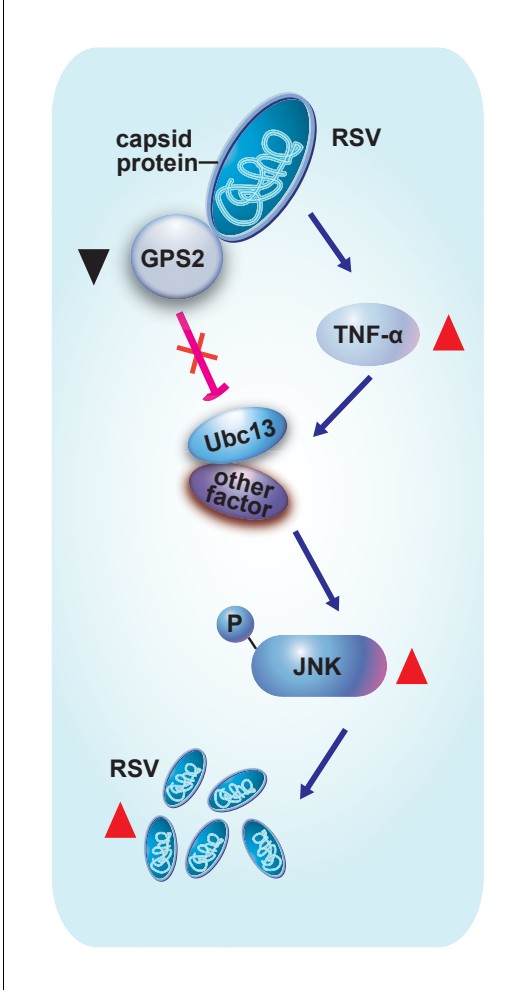

**Figure 7.** Model of RSV regulation of the JNK signaling pathway of its vector insect causing increased replication. The virus activates the JNK signaling pathway of the vector insects in three ways: i) upregulating TNF-α; ii) decreasing the expression of GPS2, which is a repressor of JNK activation; and iii) binding of the virus capsid protein to GPS2 to prevent it from inhibiting the JNK activation machinery. JNK activation is beneficial to the ability of the virus to replicate in its vector insects.

coiled-coil region of GPS2 is evidently key for the binding of CP. This explains the moderate strength of the interaction in the yeast two-hybrid assay between CP and the partial GPS2 that lacks the intact coiled-coil region. The N-terminal coiled-coil region of GPS2 is also the binding site of planthopper Ubc13. We proved that the planthopper Ubc13 is indispensable for JNK activation, as is the mammal homolog. Binding of GPS2 to Ubc13 leads to the inhibition of JNK activation. Thus the presence of CP that competitively binds GPS2 activates the JNK pathway. On the other hand, we do not exclude the possibility that other viral proteins could also be involved in the JNK regulation. A study on human T-cell lymphotrophic virus (*Jin et al., 1997*) is of particular interest in light of our results. *Jin et al. (1997)* found that this virus' protein Tax, a Type-I oncoprotein, binds to GPS2. Given our results on the binding of the RSV capsid protein to GPS2, it is possible that Tax's binding to GPS2 activates JNK. They also showed that induction of Tax expression reduced the level of GPS2 in Jurkat cells.

In the present study, we found that JNK activation by RSV is caused by increased levels of the proinflammatory signaling molecule, TNF-α. When the expression of *TNF-α* was lowered, RSV did not upregulate the phosphorylation of JNK. In mammals, TNF-α is an upstream JNK-stimulating signal that regulates a wide range of physiological activities, such as immune responses and inflammation (*Grivennikov et al., 2010*). RSV capsid protein, as a surface recognition factor, induces this immune response, as does the entire virus. In mammals, excessive activation of proinflammatory signaling pathways is harmful, promoting the development of human diseases such as autoimmune disorder, neurodegeneration, and cancer (*Amor et al., 2010*; *Grivennikov et al., 2010*). However, the inflammation in the planthopper caused by RSV seems sustainable. Perhaps this is due to the rather short lifespan of the planthopper (about 40 days under optimal conditions).

Activation of the JNK signaling pathway in the planthoppers by RSV might promote stress resistance in the insects. The JNK signaling pathway serves as a molecular sensor for various stresses. The protective functions of JNK activation have been shown in oxidative stress tolerance at the cellular level and in the aging of the organism. For example, JNK signaling activity alleviates the toxic effects of reactive oxygen species and increases the lifespan of *Drosophila melanogaster* (*Wang et al., 2003*). JNK overexpression and phosphorylation increase the lifespan of *Caenorhabditis elegans* as well as its resistance to oxidative stress and heat stress (*Oh et al., 2005*). JNK signaling also protects a host against bacterial infections by promoting apoptosis or phagocytosis (*Mizutani et al., 2003a*; *Wandler and Guillemin, 2012*). Only a few studies have shown effects of RSV infection on the physiology of the planthoppers, such as reduced fecundity and progeny hatchability, accelerated development, greater body weight, and increased abundance of a yeast-like

symbiont (*Li et al., 2015*; *Wan et al., 2015*). Possible influences of RSV on the stress tolerance of the planthoppers, such as their ability to tolerate cold stress or heat stress, deserve to be further explored.

Given the effects of JNK inhibition in postponing the incidence of disease in rice plants infected with RSV, it seems reasonable to suggest that inhibition of the JNK pathway is a potential means to benefit rice agriculture. Perhaps we can generalize from our results on the knockdown of the JNK transcripts in this study to suggest that such inhibition of the JNK pathway —by lowering JNK transcript levels, by strengthening interactions with GPS2 or by weakening the effects of TNF-α or Ubc13 — could be beneficial agriculturally. Such alterations could possibly be achieved through breeding or other means of genetic modification or, especially in the case of weakening effects, by administering appropriate chemical compounds. A successful case has been reported in which genetically engineered rice plants that expressed the RSV CP showed resistance to RSV infection (*Hayakawa et al., 1992*).

In conclusion, our study has uncovered strategies through which the RSV manipulates the JNK signaling pathway of its vector insect, the small brown planthopper, for its own benefit. Inhibition of the JNK pathway of insect vectors may, more generally, provide an important means of intervening in the transmission of persistent plant viruses.

## Materials and methods

### Small brown planthopper strains

The viruliferous and non-viruliferous small brown planthopper strains used in this study were established from a field population collected in Hai'an, Jiangsu Province, China. They were reared separately on 2 cm to 3 cm seedlings of rice *Oryza sativa* L. spp. *japonica* var. *nippobare* in glass incubators in the laboratory, as described by *Zhao et al. (2016a)*. The RSV-carrying frequency of the viruliferous strain was maintained at no less than 90% through a purification selection every three months utsing dot-ELISA with the monoclonal anti-CP antibody (*Zhao et al., 2016b*).

### Yeast two-hybrid assay

Yeast two-hybrid screening was performed with the ProQuest two-hybrid system (Invitrogen, Carlsbad, CA, USA) according to the manufacturer's protocol. A whole-body cDNA library for small brown planthopper was constructed in the GAL4 activation domain of the vector pDEST22 to create prey plasmids using a CloneMiner II kit (Invitrogen). Full-length RSV CP was cloned in the GAL4 DNA binding domain of the vector pDBLeu to create pDBLeu-CP bait plasmid. MaV203 yeast cells were co-transformed with pDBLeu-CP and pDEST22-library. Positive clones were selected on the triple dropout medium (SD/–Leu/–Trp/–His) and quadruple dropout medium (SD/–Leu/–Trp/–His/–Ade). Prey plasmids were isolated from these clones for sequencing. To confirm the interaction of bait and prey proteins, we co-transformed the two plasmids into yeast strain AH109 and repeated the selection on the triple and quadruple dropout media. The prey sequences were used in a BLAST search of the small brown planthopper transcriptome (*Zhao et al., 2016a*).

### Gene cloning and sequence analysis

Highly viruliferous or non-viruliferous planthoppers in various developmental stages and six tissues (brain, salivary gland, gut, fatbody, ovary, and testicle) were ground in liquid nitrogen, and total RNA was isolated following the standard TRIzol reagent protocol (Invitrogen). The concentration and quality of total RNA were determined using a NanoDrop spectrophotometer (Thermo Scientific, Waltham, MA, USA) and by gel electrophoresis. RNA was treated using the TURBO DNA-free kit (Ambion, Austin, TX, USA) to remove genomic DNA contamination before being used for cDNA synthesis. RNA (1 μg) was reverse-transcribed to cDNA using the Superscript III First-Strand Synthesis System (Invitrogen) and random primers (Promega, Madison, WI, USA) following the manufacturer's instructions.

Based on the small brown planthopper transcriptome (*Zhao et al., 2016a*), full-length open reading frames (ORF) of the planthopper *GPS2* gene and *Ubc13* gene were amplified with the primer pairs gps2-F/gps2-R and ubc13-F/ubc13-R (*Supplementary file 1*), respectively, from a viruliferous planthopper cDNA library and sequenced. The molecular weight of encoded GPS2 protein was predicted in ExPASy (http://web.expasy.org/compute_pi/). The cellular localization of GPS2 was

predicted using PSORT II Prediction (http://psort.hgc.jp/form2.html). The phylogenetic relation of planthopper GPS2 and its paralog, GPS1 (identified from the small brown planthopper transcriptome), to those from other insect species was analyzed with the neighbor-joining method (pairwise deletion and p-distance model) using Mega 6.06 software (RRID:SCR_000667). Bootstrap analysis (1000 replicates) was applied to evaluate the internal support for the tree topology. The full-length ORF of the RSV *CP* gene (DQ299151) was amplified with the primer pair cp-F/cp-R (*Supplementary file 1*) from the viruliferous planthopper cDNA library and sequenced.

## Protein expression, purification and antibody preparation

The ORF of *GPS2* was constructed in the pET28a vector between the restriction sites NdeI-BamHI with the primer pair gps2-his-F/gps2-his-R (*Supplementary file 1*) to generate GPS2-His plasmid. Four fragments of *GPS2* (N1 [1–465], N2 [1–657], N3 [1–903], and C [886–1296]), *JNK1*, and *JNK2* were constructed in the pET28a vector between the restriction sites NcoI-SalI with the corresponding primers, gps2N-his-F/gps2N1-his-R, gps2N-his-F/gps2N2-his-R, gps2N-his-F/gps2N3-his-R, gps2C-his-F/gps2C-his-R, jnk1-his-F/jnk1-his-R, and jnk2-his-F/jnk2-his-R, respectively (*Supplementary file 1*). An N4 fragment of *GPS2* (466–885) was constructed in the pGEX-3X vector at the SmaI site with the primers gps2N4-gst-F/gps2N4-gst-R (*Supplementary file 1*) to produce N4-GST plasmid. The ORF of *CP* was constructed in the pET28a vector between the restriction sites BamHI-XhoI with the primer pair cp-his-F/cp-his-R, and also in the pGEX-3X vector between the restriction sites BamHI-SmaI with the primers cp-gst-F/cp-gst-R to generate CP-His and CP-GST plasmids, respectively (*Supplementary file 1*). The ORF of *Ubc13* was constructed in the pET28a vector between the restriction sites NcoI-SalI with primers ubc13-his-F/ubc13-his-R, and in the pGEX-3X vector at the SmaI site with the primers ubc13-gst-F/ubc13-gst-R, to generate Ubc13-His and Ubc13-GST plasmids, respectively (*Supplementary file 1*).

The recombinant plasmids of CP, Ubc13, and JNK1 were used to transform *Escherichia coli* strain BL21 (DE3) for expression. GPS2, various GPS2 fragments, and JNK2 were expressed in *E. coli* Transetta cells. After 4 hr induction with 0.4 mM isopropyl $\beta$-D-thiogalactoside at 37°C, cells were pelleted by centrifugation and sonicated for 30 min in ice water. The supernatant from the sonicated cells was used for pull-down assay or protein purification. The expressed recombinant protein CP-His was purified using Ni Sepharose (GE Healthcare, Buckinghamshire, UK) following the manufacturer's instructions and served as antigen to produce mouse anti-CP monoclonal antibody (Beijing Protein Institute Co., Ltd., Beijing, China).

## His-tag pull down, co-immunoprecipitation and Western blotting assay

*Escherichia coli*-expressed His-tagged recombinant proteins were bound to Ni Sepharose (GE Healthcare) for 1 hr at 4°C. Then the GST-tagged recombinant proteins were added and incubated for 2 hr at 4°C. After washing with lysis buffer (20 mM sodium phosphate containing 50 mM imidazole, pH 7.4), proteins were released with elution buffer (20 mM sodium phosphate containing 250 mM imidazole, pH 7.4), and separated by SDS-PAGE gel electrophoresis. The presence of target proteins was verified by western blotting with an anti-His monoclonal antibody (CWBiotech, Beijing, China), an anti-GST polyclonal antibody (CWBiotech), or an anti-CP monoclonal antibody. Expression products from pET28a vector and pGEX-3X vector were used as negative controls.

For experiments with His-tag pull-down from insects, recombinantly expressed GPS2-His was bound to Ni Sepharose. CP-GST or GST was added and incubated for 2 hr at 4°C. Then the total proteins extracted from non-viruliferous planthoppers using PBS buffer (pH 7.4) were added and incubated for another 2 hr at 4°C. In another experiment, after the recombinantly expressed GPS2-His was bound to Ni Sepharose, the total proteins extracted from non-viruliferous or viruliferous planthoppers using PBS buffer (pH 7.4) were added and incubated for 2 hr at 4°C. After washing with lysis buffer (20 mM sodium phosphate containing 50 mM imidazole, pH 7.4), target proteins were collected with elution buffer (20 mM sodium phosphate containing 250 mM imidazole, pH 7.4).

For co-immunoprecipitation, total protein was extracted from about 20 mg of the fourth instar non-viruliferous planthoppers using T-PER Tissue Protein Extraction Reagent, containing a protease inhibitor cocktail (Thermo Fisher Scientific). Around 10% of total extracted protein was reserved as input for further western blot analyses. The remaining protein was cleared with 10 μL of Dynabeads

Protein G (Novex by Thermo Fisher Scientific). 40 µL of Protein G beads were prepared with 5 µg of CP monoclonal antibody bound, and then incubated with 200 µL of CP-His protein for 30 min. The mouse IgG (Merck Millpore, Billerica, MA, USA) was used as negative control. 200 µL of cleared total protein of planthoppers were then immunoprecipitated with the bead–antibody–CP complex for 30 min. The antibody–CP-GPS2 complex was dissociated from the beads with elution buffer (Novex by Thermo Fisher Scientific) for western blot analysis.

Planthopper GPS2, Ubc13, and CP proteins in insects were recognized by an anti-human GPS2 goat polyclonal antibody (Santa Cruz Biotechnology, Dallas, TX, USA; RRID:AB_10841240), an anti-human Ubc13 mouse monoclonal antibody (Santa Cruz Biotechnology; RRID:AB_11150503), and an anti-CP monoclonal antibody, respectively, in western blotting. 10 mg of insects were used in all western blotting analyses.

For JNK detection, total protein was extracted using RIPA lysis buffer, containing protease inhibitor cocktail and phosphatase inhibitor cocktail (CWBiotech). Total JNK, phosphorylated JNK, and tubulin in various planthopper samples were detected using an anti-human JNK2 polyclonal antibody (Cell Signaling Technology, Danvers, MA, USA; RRID:AB_2250373), an anti-phospho-human polyclonal JNK2 antibody (Cell Signaling Technology; RRID:AB_331659), and an anti-human tubulin monoclonal antibody (CWBiotech), respectively, in western blotting. For another group of experiments, total protein was extracted using RIPA lysis buffer and incubated with λ-phosphatase (New England Biolabs, Ipswich, MA, USA) for 1 hr at 30°C. After the treatment, total JNK and phosphorylated JNK were detected by western blotting. The density of phosphorylated JNK was quantified with image analysis software ImageJ and normalized to that of tubulin. Differences were statistically evaluated using Student's $t$-test in SPSS 17.0 (RRID:SCR_002865).

Proteins were isolated from nuclei and cytoplasm of insect whole body using the Nuclear and Cytoplasmic Protein Extraction Kits (Beyotime, Jiangsu, China) according to manufacturer's instructions. The reference proteins for nuclear and cytoplasmic proteins were histone H3 and tubulin, respectively, which were displayed using an anti-human H3 monoclonal antibody (Beijing Biodragon Immunotechnologies, Beijing, China) and an anti-human tubulin monoclonal antibody (CWBiotech).

## Immunofluorescence microscopy

Salivary glands were dissected from non-viruliferous four-instar planthopper nymphs in cold distilled water on a glass plate, and fixed in 4% paraformaldehyde for 2 hr at room temperature. After being permeabilized with osmotic buffer (0.01 M phosphate-buffered saline containing 2% Triton X-100, pH 7.4) for 4 hr, the salivary glands were blocked with 1% bovine serum albumin for 30 min at room temperature. The samples were incubated with the primary antibody, anti-human GPS2 goat polyclonal antibody (Santa Cruz Biotechnology), overnight at 4°C. After washing with 0.01 M phosphate-buffered saline containing 1% Tween-20 (pH 7.4), the secondary antibody, Alexa Fluor 594 (red) affinipure donkey anti-goat IgG (YEASEN, Shanghai, China), was added. The nuclei were counterstained with Hoechst (blue) in accordance with the manufacturer's instructions (Invitrogen). Negative control was without the primary antibody. The images were viewed under a Leica TCS SP5 confocal microscope (Leica Microsystems, Solms, Germany). Twenty salivary glands were tested.

## Feeding small brown planthoppers RSV crude preparations

Non-viruliferous four-instar nymphs were fed on an artificial diet containing RSV crude preparations from RSV-infected rice seedlings for 8 hr as previously described (*Zhao et al., 2016b*) and then transferred to healthy rice seedlings.

## Double-stranded RNA synthesis and delivery

PCR primers with T7 promoter sequences, gps2-dsRNA-F/gps2-dsRNA-R, jnk1-dsRNA-F/jnk1-dsRNA-R, jnk2-dsRNA-F/jnk2-dsRNA-R, ubc13-dsRNA-F/ubc13-dsRNA-R, or TNF-α-dsRNA-F/ TNF-α-dsRNA-R were used to prepare 219-bp double-stranded RNA (dsRNA) of *GPS2*, 124-bp dsRNA of *JNK1*, 167-bp dsRNA of *JNK2*, 142-bp dsRNA of *Ubc13*, or 107-bp dsRNA of *TNF-α* (*Supplementary file 1*). A 420-bp dsRNA for green fluorescent protein (GFP) was amplified using primers gfp-dsRNA-F and gfp-dsRNA-R as negative controls (*Supplementary file 1*). dsRNA was

generated using the T7 RiboMAX Express RNAi System (Promega, Madison, Wisconsin, USA) and purified using Wizard SV Gel and the PCR Clean-Up System (Promega) following the manufacturers' protocols. Injection of 23 nL of dsRNAs at 6 µg/µL was performed on the fourth instar nymphs. The dsRNAs were delivered into hemolymph in the ventral thorax by microinjection through a glass needle using Nanoliter 2000 (World Precision Instruments, Sarasota, Florida, USA).

## Quantitative real-time PCR

Quantitative real-time PCR (qRT-PCR) was used to quantify the relative RNA levels of RSV *CP* and the transcript levels of *GPS2*, *JNKs*, *TNF-α*, and *Ubc13* in extracts of whole body or various tissues of planthoppers. A 172-bp fragment of *CP*, a 148-bp fragment of *GPS2*, a 118-bp fragment of *JNK1*, a 110-bp fragment of *JNK2*, a 100-bp fragment of *Ubc13*, and a 100-bp fragment of *TNF-α* were amplified using the primer pairs cp-q-F/cp-q-R, gps2-q-F/gps2-q-R, jnk1-q-F/jnk1-q-R, jnk2-q-F/jnk2-q-R, ubc13-q-F/ubc13-q-R, and TNF-α-q-F/TNF-α-q-R, respectively (*Supplementary file 1*). qRT-PCR was carried out in 20 µl of reaction agent composed of 2.5 µl of template cDNA, 10 µl of 2×SYBR Green PCR Master Mix (Fermentas, Waltham, MA, USA), and 0.25 µM each primer on Light Cycler 480 II (Roche, Basel, Switzerland). The thermal cycling conditions were 95°C for 2 min, followed by 40 cycles of 95°C for 30 s, 60°C for 30 s and 68°C for 40 s. The transcript level of planthopper translation elongation factor 2 (*ef2*) was quantified with primer pair ef2-q-F/ef2-q-R to normalize the cDNA templates of planthoppers (*Supplementary file 1*). The relative transcript level of each gene was reported as mean ± SE. Differences were statistically evaluated using SPSS 17.0. Student's t-test was performed to compare two means, whereas one-way ANOVA followed by a Tukey's test was applied for multiple comparisons.

## Quantification of *GPS2* expression in various tissues and developmental stages of viruliferous and non-viruliferous planthoppers

Six tissues (brain, salivary glands, digestive gut, fat body, ovary, and testicle) were collected from 20 to 50 viruliferous and non-viruliferous adult planthoppers for RNA extraction. Six replicates for each tissue were prepared. RNA was also isolated from the eggs, the first to fifth instars, and female and male adults of non-viruliferous planthoppers, and from the fourth instars, and female and male adults of viruliferous planthoppers. Six replicates and six to ten instars or adults, and 30 eggs per replicate were prepared for each developmental stage. The non-viruliferous fourth instars were fed on the artificial diet containing RSV crude preparations for 8 hr and then raised on healthy rice seedlings for from one to five days. The insects whose food did not contain RSV were used as controls. Five replicates and six insects per replicate from each day were prepared for RNA isolation. The expression levels of *GPS2* were quantified using qRT-PCR.

## Injection of CP protein in insects

Recombinantly expressed and purified CP-His was micro-injected into the hemolymph of non-viruliferous fourth instars in the ventral thorax using Nanoliter 2000 (World Precision Instruments). For each insect from the treatment group, 23 nL of CP-His at 330 µg/mL was injected. An equal volume of the expression and purified products from the pET28a vector was injected into the insects from the control group. Activation of JNK, accumulation of TNF-α, and *GPS2* transcript levels were checked at 24 hr after CP-His administration. Three to six biological replicates and eight insects per replicate were used.

## Injection of *Ubc13* dsRNA and TNF-α

Non-viruliferous fourth instar planthoppers were injected with 23 nL of dsRNA of *Ubc13* at 6 µg/µL and with 23 nL of 50 µM mouse TNF-α (Sigma-Aldrich, Santa Clara, CA, USA) using Nanoliter 2000 (World Precision Instruments). The insects from the control group were injected with 23 nL of dsRNA of *GFP* at 6 µg/µL and 23 nL of 20 mM Tris-HCl containing 0.1% Tween 20. *Ubc13* transcript levels and activation of JNK were checked after 3 d of treatment. At least three biological replicates and eight insects per replicate were used.

### Enzyme linked immunosorbent assay (ELISA) on TNF-α

Eight four-instar nymphs from viruliferous or non-viruliferous planthopper strains, or from CP-His injected non-viruliferous planthoppers or negative control insects, were ground in 200 μL of 0.01M PBS buffer (pH 7.2) containing protease inhibitor cocktail (CWBIO, Beijing, China). The supernatant was collected after centrifugation at 12,000 g for 15 min at 4°C and the protein concentration was determined using the Bradford method. The TNF-α concentration from each sample of planthoppers was determined using a human TNF-α ElISA Kit (Liuhe, Wuhan, China), in which the monoclonal antibody of human TNF-α and a standard curve of human TNF-α with a two-fold dilution from 1000 pg/ml to 15.6 pg/ml were supplied. The quantity of TNF-α from each sample was designated as pg of TNF-α per mg of total protein. Six (viruliferous versus non-viruliferous planthoppers) or eight (CP-His injected experiment) biological replicates and eight fourth instars per replicate were used for each group.

### Quantification of RSV proliferation in planthoppers

Non-viruliferous fourth instar planthoppers were fed an artificial diet containing RSV crude preparations for 8 hr, and then raised on healthy rice seedlings. After 2 d, the insects were injected with 23 nL of dsRNA mixture of JNK1 and JNK2 at 6 μg/μL, or dsRNA of TNF-α at 6 μg/μL, or 0.9 μM JNK specific chemical inhibitor SP600125 (Sigma-Aldrich), or 50 μM mouse TNF-α (Sigma-Aldrich), and then raised on healthy rice seedlings for 3 d. For GPS2 knockdown, insects were injected with 23 nL of dsRNA of GPS2 at 6 μg/μL. Two days later the insects were fed artificial diet containing RSV crude preparations for 8 hr and then raised on healthy rice seedlings for 5 d. The transcript knockdown of JNKs, TNF-α, GPS2 and the RNA level of CP were measured using qRT-PCR. Insects injected with 23 nL of dsGFP-RNA at 6 μg/μL were used as controls. Control groups for SP600125 and TNF-α were injected with 23 nL of 2% DMSO or 20 mM Tris-HCl containing 0.1% Tween 20, respectively. Six biological replicates and four insects per replicate were used for the dsRNA injection experiment and the pharmacological treatment experiment.

### Disease incidence rates of rice plants when GPS2 or JNKs were knocked down

After RSV was incubated in the GPS2 or JNKs knocked down planthoppers for 5 d, five insects were transferred to two new healthy rice seedlings for 24 hr and then removed from the plants. The plants were then placed in a greenhouse at 20°C or 26°C with 16 hr of light per day to allow observation of disease symptoms. Six groups of plants per replicate and four or five replicates were used to calculate the disease incidence rates on each day. Insects injected with 23 nL of dsGFP-RNA at 6 μg/μL were used as controls.

## Acknowledgements

We thank Prof. Rongxiang Fang's group from the Institute of Microbiology, Chinese Academy of Sciences for providing the cDNA library of small brown planthopper for yeast two-hybrid screening. We also thank Prof. Gerald Reeck from Kansas State University for comments and language suggestions. This work was supported by grants from the Strategic Priority Research Program of the Chinese Academy of Sciences (No. XDB11040200), the Major State Basic Research Development Program of China (973 Program) (No. 2014CB13840402), and the Natural Science Foundation of China (No. 31772162), and by a grant from the State Key Laboratory of Integrated Management of Pest Insects and Rodents (No. ChineseIPM1604).

## Additional information

### Funding

| Funder | Grant reference number | Author |
|---|---|---|
| Chinese Academy of Sciences | Strategic Priority Research Program XDB11040200 | Feng Cui |
| Ministry of Science and Tech- | Major State Basic Research | Feng Cui |

| nology of the People's Republic of China | Development Program of China 2014CB13840402 | |
| --- | --- | --- |
| National Natural Science Foundation of China | 31772162 | Le Kang |
| Chinese Academy of Sciences | Grant of State Key Laboratory of Integrated Management of Pest Insects and Rodents ChineseIPM1604 | Feng Cui |

The funders had no role in study design, data collection and interpretation, or the decision to submit the work for publication.

### Author contributions

WW, WZ, JL, LL, Investigation; LK, Conceptualization, Supervision, Writing—review and editing; FC, Conceptualization, Supervision, Writing—original draft, Writing—review and editing

### Author ORCIDs

Le Kang, http://orcid.org/0000-0003-4262-2329
Feng Cui, http://orcid.org/0000-0001-6215-7159

## Additional files

### Supplementary files

• Supplementary file 1. Primers used in this study.

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
