## [Decision Letter]

Thank you for submitting your article "The JNK pathway of a vector insect is activated by virus capsid protein and promotes viral replication" for consideration by *eLife*. Your article has been reviewed by three peer reviewers, and the evaluation has been overseen by a Reviewing Editor and Wenhui Li as the Senior Editor. The following individual involved in review of your submission has agreed to reveal her identity: Isgouhi Kaloshian (Reviewer #3).

The reviewers have discussed the reviews with one another and the Reviewing Editor has drafted this decision to help you prepare a revised submission.

Summary:

Little was known about how insect-borne viruses manipulate the cellular JNK signaling pathway of their vector insects. In this manuscript, the authors studied the response of the JNK pathway in the vector insect, the small brown planthopper, to infection by the plant virus RSV. They found that the planthopper increased its level of TNF-α and decreased its level of G protein Pathway Suppressor 2 (GSP2) after RSV infection. The virus capsid protein competitively bound GPS2 to release it from inhibiting the JNK activation machinery. They further showed that JNK activation promoted RSV replication in the insect vector, while JNK inhibition reduced virus production and postponed the plant morbidity. Overall, the findings are interesting and potentially important. The experiments are well executed and most conclusions are well supported by the data presented in this work. This manuscript is suitable for publication in *eLife* after appropriate revisions.

Essential revisions:

1) In light of the different patterns of JNK and P-JNK bands seen in Figure 3 and Figure 4 (see details below), it is critical that expression of these two JNK genes and the specificity of the anti-human JNK and anti-human P-JNK antibodies used in this study be verified. For example, in Figure 3, two major bands were detected by the anti-JNK antibody in the "Non-viruliferous insects" sample (right, middle panel), similar to what was observed in Figure 3, however, only one major band was detected by the anti-JNK antibody in the "Viruliferous insects" sample (left, middle panel). Why? Moreover, because these samples were not run on the same blot, it is difficult to directly compare the mobility of the bands. Along the same line, one or two bands were detected by the anti-P-JNK antibody in different samples: e.g. one band in "control" sample (Figure 3, top panel; possibly P-JNK1 as suggested by comparing it to the MW markers), one band in "Salivery gland, viruliferous" sample (Figure 4; is this the same band as in Figure 3, "control", top panel?), two major bands in "Ovary, viruliferous" sample (Figure 4; no MW markers indicated on the blot – could they correspond to P-JNK1 and P-JNK2?), one major and slower migrating band plus one minor and faster migrating band in "Testis, viruliferous" sample (Figure 4). Similarly, there were one minor and slower migrating band plus one major and faster migrating band in "Female" samples, but only one major band in "Male" samples (Figure 4). Please clarify these potential sources of confusion.

2) For all figures attempting to show increase or decrease in JNK activation (mostly in Figure 3 and Figure 4), quantitative analysis should be performed by densitometry of P-JNK image bands, with normalizing for total JNK and for protein loading (tubulin) in the same sample.

3) Figure 4: Based on data from Figure 3, decreased level of GPS2 should be associated with increased level of P-JNK. However, this is obviously not the case if lanes 2 and 3 were visually inspected. Again, quantitative analysis is required to provide more objective conclusions.

---

## [Author Response]

*Essential revisions:*

*1) In light of the different patterns of JNK and P-JNK bands seen in Figure 3 and Figure 4 (see details below), it is critical that expression of these two JNK genes and the specificity of the anti-human JNK and anti-human P-JNK antibodies used in this study be verified. For example, in Figure 3, two major bands were detected by the anti-JNK antibody in the "Non-viruliferous insects" sample (right, middle panel), similar to what was observed in Figure 3, however, only one major band was detected by the anti-JNK antibody in the "Viruliferous insects" sample (left, middle panel). Why? Moreover, because these samples were not run on the same blot, it is difficult to directly compare the mobility of the bands. Along the same line, one or two bands were detected by the anti-P-JNK antibody in different samples: e.g. one band in "control" sample (Figure 3, top panel; possibly P-JNK1 as suggested by comparing it to the MW markers), one band in "Salivery gland, viruliferous" sample (Figure 4; is this the same band as in Figure 3, "control", top panel?), two major bands in "Ovary, viruliferous" sample (Figure 4; no MW markers indicated on the blot – could they correspond to P-JNK1 and P-JNK2?), one major and slower migrating band plus one minor and faster migrating band in "Testis, viruliferous" sample (Figure 4). Similarly, there were one minor and slower migrating band plus one major and faster migrating band in "Female" samples, but only one major band in "Male" samples (Figure 4). Please clarify these potential sources of confusion.*

Thanks for this suggestion. We expressed the two JNK proteins of small brown planthoppers in *E. coli* and tested the specificity of the anti-human JNK2 polyclonal antibody. The results are presented in a new figure, Figure 3—figure supplement 2. We found that the anti-human JNK2 antibody can recognize both of the two planthopper JNKs. We complement this information in the manuscript: The recombinantly expressed planthopper JNK1 and JNK2 were recognized by the anti-human JNK2 polyclonal antibody (Figure 3—figure supplement 2) (subsection “GPS2 represses JNK activation in the planthoppers”). As for the one or two bands detected by the anti-JNK or anti-P-JNK antibody in planthopper samples, we think that there are differences between the two JNKs in their expression levels in different organs. But this does not influence the main conclusion of this work.

*2) For all figures attempting to show increase or decrease in JNK activation (mostly in Figure 3 and Figure 4), quantitative analysis should be performed by densitometry of P-JNK image bands, with normalizing for total JNK and for protein loading (tubulin) in the same sample.*

We performed the densitometry analysis for P-JNK and GPS2 image bands from Figure 2 to Figure 5. The relative densities of P-JNK and GPS2 were normalized with those of tubulin and presented as mean ± SE. The analysis method is supplemented in subsection “His-tag pull down, co-immunoprecipitation and Western blotting assay” The results are supplied as respective figure supplements.

*3) Figure 4: Based on data from Figure 3, decreased level of GPS2 should be associated with increased level of P-JNK. However, this is obviously not the case if lanes 2 and 3 were visually inspected. Again, quantitative analysis is required to provide more objective conclusions.*

We performed the quantitative analysis on the density of P-JNK and GPS2 image bands compared to that of tubulin. A significant increase of P-JNK and a significant decrease of GPS2 were confirmed (Figure 4—figure supplement 1).